# Peptide Specific Nanoplastic Detection Based on Sandwich Typed Localized Surface Plasmon Resonance

**DOI:** 10.3390/nano11112887

**Published:** 2021-10-28

**Authors:** Seungju Oh, Hyeyeon Hur, Yoonjae Kim, Seongcheol Shin, Hyunjeong Woo, Jonghoon Choi, Hyun Ho Lee

**Affiliations:** 1Department of Chemical Engineering, Myongji University, Yongin-Si 17058, Korea; sjtleo55@gmail.com (S.O.); hhuh9801@gmail.com (H.H.); yse459@gmail.com (Y.K.); lms4278@gmail.com (S.S.); 2Department of Biomedical Engineering, School of Integrative Engineering, Chung-Ang University, Seoul 06974, Korea; hyunjeong1226@gmail.com

**Keywords:** nanoplastics, LSPR, peptide binding, sandwich assay

## Abstract

Recently, various waste microplastics sensors have been introduced in response to environmental and biological hazards posed by waste microplastics. In particular, the detrimental effects of nano-sized plastics or nanoplastics have been reported to be severe. Moreover, there have been many difficulties for sensing microplastics due to the limited methodologies for selectively recognizing nanoplastics. In this study, a customized gold nanoparticles (Au NPs) based localized surface plasmon resonance (LSPR) system having bio-mimicked peptide probes toward the nanoplastics was demonstrated. The specific determination through the oligo-peptide recognition was accomplished by chemical conjugation both on the LSPR chip’s 40~50 nm Au NPs and sandwiched 5 nm Au NPs, respectively. The peptide probe could selectively bind to polystyrene (PS) nanoplastics in the forms of fragmented debris by cryo-grinding. A simple UV-Vis spectrophotometer was used to identify the LSPR sensing by primarily measuring the absorbance change and shift of absorption peak. The sandwich-binding could increase the LSPR detection sensitivity up to 60% due to consecutive plasmonic effects. In addition, microwave-boiled DI water inside of a styrofoam container was tested for putative PS nanoplastics resource as a real accessible sample. The LSPR system could be a novel protocol overcoming the limitations from conventional nanoplastic detection.

## 1. Introduction

Recently, the fragmented microplastics of waste products used by humans on a daily basis have been receiving attention as a serious threat to biological systems and to the global environment [1,2,3]. Microplastics are typically defined as plastics with sizes ranging from 5 mm to a few nanometers. Particularly, the most harmful microplastics are typically known to have a size smaller than 1 mm, and have the potential to bind toxic substances, which are likely to be mistaken for food by marine animals [4,5,6].

Moreover, within the broader category of microplastics, nano-sized microplastics, or nanoplastics, have been reported to be critically hazardous to human beings. However, the effectiveness of conventional instrumental analysis methods has been limited because of the extremely small size of nanoplastics (<100 nm) [7,8].

The global environmental presence of microplastics mandates that ecological/biological risk assessments effectively detect and quantify the smallest nanoplastics. In addition, the high diversity of micro/nanoplastics in terms of their various sizes, shapes, colors, and polymer type make data comparison more difficult for researchers, as they have yet to reach a consensus on a micro/nanoplastics classification protocol. In fact, many researchers who have employed centrifugation and/or sieving methods to collect nanoplastics have reported various buoyant force- and density-related difficulties [9,10,11].

In addition to the difficulties associated with the nanoplastics sampling procedures, there are various additional problems with nanoplastic (and microplastic) detection systems, as these systems require a staining protocol for fluorescent detection, or massive optical detection systems that entail the use of Fourier transform infrared (FTIR) or Raman spectroscopy [12,13,14,15].

The earliest and simplest method of microplastic analysis was visual identification through the use of an optical microscope; however, the error rate associated with this method of analysis tends to be very high, i.e., in the 20% to 70% range. Maes et al. proposed a staining method in which the fluorescent dye Nile Red is employed [16]. Their method entails placing fine plastic particles in a sample, and subsequently applying blue light and orange filters to fluoresce them under a microscope. However, because this method has not yet been tested on complex samples containing organic matter, the dye attachment may be non-selective.

For fluorescent staining, there have been huge biases with shape (e.g., sphere) and plastic species about microplastics especially less than 300 μm [17]. Therefore, if shape or morphology independent proving strategy toward microplastics could be proposed and adopted, its application will be more efficient than the fluorescent methods.

Meanwhile, oligo-peptides have been conceptually mimicked by receptors from biological membranes due to their specific binding affinities [18,19]. In fact, there have been several reports about the screening of oligo-peptides, which can selectively bind to the surfaces of microplastics according to their species, e.g., polystyrene (PS), polyethylene (PE), and polypropylene (PP) [18,19].

Among the various microplastic-specific binding peptides that have been reported to date, relatively short peptides, such as the PS-targeted oligo-peptides, are the most common [20,21,22]. Specifically, there are very short peptides with only seven amino acids, i.e., HWGMWSY (histidine-tryptophan-glycine-methionine-tryptopahn-serine-tyrosine), which is PS binding targeted.

Here, if the microplastic-specific peptides were utilized for the detection portfolio, the concerns with respect to the shape and morphology would be diminished. The development of photo-analysis technology that detects microplastics via a peptide-binding method is expected to increase the accuracy and simplify the process of microplastic detection.

Localized surface plasmon resonance (LSPR) is generated by a light wave trapped within conductive nanoparticles smaller than the wavelength of light [23]. The binding of a biological molecule to the surface of metallic nanoparticles changes the local refractive index and, consequently, induces a shift in the LSPR wavelength. Examples of using LSPR technology to accurately identify various biomarkers have been reported. In this study, we employed a peptide-binding method to capture microplastics; then, we developed a system to sort the microplastics using LSPR sensors [24,25,26,27].

Simple LSPR-based systems that employ a sensor chip with gold (Au) nanoparticles (NPs) have recently been demonstrated to be effective in its sensitivity and reusability [28,29]. Especially, short length oligo-peptide related to bioreceptor has been successfully utilized for the efficient detection of target in the LSPR format [28]. Because the PS is an insulator and do not conduct electricity, it is highly unlikely that polarizing charges will be found to exist at the surface of the microplastics. Thus, LSPR sensitivity will be low. Here, additional 5 nm Au NPs were supplied to ensure that Au NPs could attach to the other side of the microplastics to increase sensitivity. In fact, the sandwich typed assay employing consecutive bindings have been introduced for biomolecular assays [30]. Therefore, to increase selectivity, the PSBP was conjugated to Au NPs of two different sizes, i.e., 40~50 nm Au NPs on the LSPR chip, and 5 nm for sandwich binding; then, the existence of PS was confirmed by attaching the two differently sized Au NPs to create a sandwich assay.

In this study, we report that LSPR chips with additional sandwich binding of plastic specific oligo-peptide tethered 5 nm Au NPs could successfully detect nanoplastics fragmented from grinded microplastics, which were not conveniently synthesized spherical nanoplastics as model samples. The bindings of various morphological nanoplastics on the LSPR sensor were proved and validated through field emission scanning electron microscopy (FE-SEM) analysis. Therefore, the LSPR system in this study could be an efficient tool for a simple measurement of nanoplastics without restrictions of their shape, transparency, and morphology.

## 2. Materials and Methods

### 2.1. LSPR Sensor Chip

Au NP layered sensor chips (Plexense Co., Yongin, Korea) were purchased for implementation in the UV-Vis spectrophotometer measurement system [28,29]. The sensor chip consists of Au NPs (diameter: 40~50 nm) that form a monolayer on a poly carbonate (PC) plastic sensor plate template; this monolayer is shown as the red layer in Figure 1.

Accordingly, individual four PC plates with 40~50 nm Au NPs were constructed to realize the reliable and efficient measurement of the absorbance signal. In this format, sensing probe molecules can be conjugated on the 40~50 nm Au NPs layers that induce LSPR effect or change extinction coefficient of the Au NPs layers [28,29].

### 2.2. Plastic-Specific Binding Oligo-Peptides and Conjugation on Au NPs

A specific PSBP was used to selectively capture PS microplastics, which were conjugated to 40~50 nm Au NPs on an LSPR chip, as well as 5 nm Au NPs. The PS-targeted peptide or PS binding peptide (PSBP) used in this study, i.e., HWGMWSY, consists of short peptides with seven amino acids. The oligo-peptide was functionalized at an N-terminal with a thiol (−SH) group to be tethered to 40~50 nm and 5 nm Au NPs, respectively.

Because the oligo-peptide powders could not be directly dissolved in DI water, each type of plastics binding peptides was dissolved into dimethylsulfoxide (DMSO) at a concentration of 2 mg/mL, stored as a stock solution at 4 °C by addition of DI water, it was diluted to 0.4 mg/mL for conjugation of Au NPs.

### 2.3. Preparation of PS Nanoplastics

A grinding or cryo-grinding process was employed for the preparation of microplastic fragments or debris. After the grinding, the plastic powder was applied into the sieve and desired sized fraction was harvested by the mesh size of separation sieve. It was confirmed by taking a photomicrograph to determine whether it was properly separated in the relevant size section. The size of the PS fragment was photographed through the image J program and a size distribution could be drawn (Appendix A). PS solutions with different concentrations were created by diluting under 10 μm sized PS and 75~106 μm sized PS particles in DI water; note that the PS was suspended without other additives. The final PS solutions were prepared at concentrations of 0.001, 0.01, 0.1, 1.0, and 10 mg/mL, respectively. Each sample solution had nanoplastic portion as the debris during cryo-grinding process.

### 2.4. LSPR Chip Measurement and SEM Analysis

All stock solutions for the PSBP was prepared at a concentration of 2 mg/mL via dissolution with DMSO. A 40~50 nm Au coated LSPR chip (Plexense. Co, Yongin-si, Korea) was thiol-conjugated by immersing it in the PSBP stock solution at 10× dilution (0.2 mg/mL) for 20 min. Then, the chip was immersed into the PS sample solution for 20 min to allow time for the chip to attach to the PS nanoplastic samples. The LSPR effect was analyzed to find out different absorbance intensity and wavelength shift by UV-Vis spectrophotometer (Helios Alpha, Thermo Scientific, Alva, UK; UV-1601, Shimazu, Kyoto, Japan). The scanning rate of UV-Vis spectroscopy was set to 3800 nm/min. For the scanning electron microscopy (SEM) analysis, using a field emission scanning electron microscopy (FE-SEM, SU-70, Hitachi Co., Osaka, Japan), surface of LSPR chip was analyzed. Because the LSPR chip was a nonconductive target, Au was used as a coating layer to enable SEM measurement. In addition, from a bench-top SEM (COXEM, EM-30AX, Daejon, Korea), microplastics dried on glass surface could be imaged with Au coating.

### 2.5. Conjugation of PSBP to 5 nm Au NPs and Electrophoresis for Sandwich Assay

To increase the LSPR sensitivity to micro/nanoscale PS, Au NPs were added on both sides of the PS nanoplastics, which were already bound to the 40~50 nm Au NP layer of the LSPR chip.

The PSBP and 5 nm Au NPs conjugation procedure is as follows: Incubate a mixture of 2 mL of 5 nm Au NPs (Ted Pela, Co., Redding, CA, USA) and PSBP (50 μg/mL) solution for 20 min to conjugate the thiol-functionalized PSBP. Then, use an ultrafiltration kit (Vivaspin 500, Pall Co., Port Washington, NY, USA) to perform centrifugal separation until quarter of them are left. The filtration process serves the purpose of removing the non-conjugated PSBP on the LSPR chip surface. Dilute this solution with DI water two times before injecting it into the PS-conjugated chip for 20 min to combine the PSBP on multiple sides of the PS. To examine degree of conjugation of PSBP to 5 nm Au NPs, agarose gel (1%, sigma-Aldrich, St. Louis, MO, USA) electrophoresis was performed in 1 × SB buffer (1 mM sodium borate, sigma-Aldrich) under 100 V in electrophoresis kit (Biorad Co., Hercules, CA, USA) Ultra-centrifugation tube (Viva spin 500) was used to centrifugate 5 nm Au NPs gel to filter out the unbound PSBP. The remained suspension was put it in the well of agarose gel, and electrical field of 100 V was applied.

### 2.6. PDMS Microfluidic Trap Module for Nanoplastic Collection

PDMS (Sylgard 184, Dow corning, Midland, MI, USA) microfluidic substrate was made by mixing with polymeric PDMS solution (DC-184A) and hardener (DC-184B) for 10:1 ratio. The PDMS pre-solutions were poured on flat surface and the film was put to vacuum desiccator to remove any remained bubbles completely. It was performed in atmosphere for 30 min and in another vacuum state for 30 min. Then, the film was heated and crosslinked in a dry oven at 72 °C for two hours before laser processing. To fabricate microfluidic trap module directly, the hardened PDMS layer was shaped by laser ablation. The programmable CO_2_ laser (Infrared light) writer (PL-40K, Korea stamp, Seoul, Korea) was used to make the trap channel on PDMS plate. The laser beam was focused with a lens (*f* = 30 mm) onto the PDMS surface. The programmable laser writer could change its power at 40~50 Watts generating thermal energy. The PDMS microfluidic module for nanoplastic trapping were constructed to have 77 mm long, 10 mm wide, and 0.5 mm high channel having nine alternatively positioned traps (2.0 × 3.5 mm).

### 2.7. Styrofoam Originated PS Nanoplastic Sample Preparation

For analysis of PS nanoplastics boiled in styrofoam plastic containers, 200 mL DI water at room temperature was poured in a styrofoam container and the ID water was boiled in microwave. For recirculation of nanoplastic sample solution toward the microfluidic trap module, 2 mm OD silicon tubings were connected at each side of PDMS microfluidic and peristaltic pump was applied with 0.1 mg/mL flow rate for 30 min to concentrate PS nanoplastics inside of traps.

## 3. Results and Discussions

An image of the oligo-peptide used in this study is shown in Figure 1. Three dimensional structure of the PSBP was drawn through PEP-FOLD 3 (De novo peptide structure prediction). Available online: https://bioserv.rpbs.univ-paris-diderot.fr/services/PEP-FOLD3/ (26 October 2021). The Figure 1 also shows a diagram of the LSPR chip used system in this study. The 40~50 nm Au NPs layers in the customized LSPR chip were coupled by PSBP at first. Then, fragmented PS samples were applied to be specifically recognized and captured by the immobilized PSBP on the Au NPs layer. To impose consecutive LSPR phenomena or sandwich assay effect, additional PSBP coupled 5 nm Au NPs were applied.

As it was initially believed that efficiency of the LSPR-based system could be determined by evaluating selective binding toward the nanoplastics, absorbance change due to binding between PSBP and PS nanoplastics were examined for different concentrations of PS. As shown in Figure 2, specific binding between PSBP and PS could cause increased absorbance of UV-Vis spectra. To ensure no effect on the baseline of UV-Vis absorption, PSBP solution in DMSO was examined in UV-Vis spectra as shown in Appendix A. As shown in Appendix A, there is no particular absorption peak obtained from the PSBP itself.

As shown in the Figure 2, the highest absorbance increase was found to be approximately by 0.12 in magnitude. It occurred when the PS concentration was at 0.1 mg/mL. The absorbance increases upon binding of PS sample in Figure 2 was not believed to be from total amount of microplastics, which were contained in PS sample solution. In other words, the reason why high concentrations of 1 mg/mL PS samples showed lower absorbance than 0.1 mg/mL in Figure 3c is that the incident light for UV-Vis spectra could be scattered in the cases of 1 mg/mL PS samples with large-sized (>1 μm) fragments. In fact, Appendix A shows the SEM measurement results for the surfaces of LSPR chip after binding of 1 mg/mL PS sample, as viewed from the (a) top and (b) side. The SEM analysis revealed that relatively large micro-sized PS debris could be easily observed because the PS fragments were clumped together on the LSPR chip surfaces to have a screen effect for UV-Vis spectral light. The SEM images obtained at 2000× to 5000× magnification could reveal that the microplastics were aggregated to form approximately up to 50 μm diameter masses. Thus, the UV-Vis absorbance was found to decrease with increase of concentration once when the PS sample concentration of 1 mg/mL was reached as like as Figure 2. For the practical application, environmental samples in water bottles could have ~100 particles/L [13]. Therefore, the concentration of 0.1 mg/mL PS sample in this study can be far beyond the microplastic concentrations of practical drinking water or food sample [12,15]. In Figure 2, the absorbance spectra with 0 mg/mL PS was a control sample. Separately, a PS sample of 0.0001 mg/mL was examined in the LSPR detection, which did not show distinguishable absorbance change from the control sample. Therefore, it was defined that a limit of quantification (LOQ) was 0.001 mg/mL PS, which could be defined as a lowest concentration of a substance that is possible to be determined by the LSPR of this study.

Nevertheless, nano-sized fragments and debris also were also observed to be attached to or scattered around the micro-sized microplastics, as shown in SEM images of Appendix A. Thus, by separate and repetitive experiments, it could prove that nano-sized plastics or nanoplastics, which were contained as debris or fragments from microplastics, were sources for the LSPR detection as represented as the absorbance rise.

It has been reported that, if the PSBP is only attached to one side of a PS particle, another side may be available for attachment to another PSBP [21,22]. Moreover, nanoplastics are not isotropically spherical. Thus, as previously reported, an extra LSPR effect or consecutive LSPR effect can generally be anticipated by sandwich method with additional PSBP conjugated 5 nm Au NPs bindings [31].

In addition, cryo-grinded polyprolylene (PP) was applied to the PSBP. There was relative binding of the PSBP to the surface of the PP fragment. However, when the PP fragment (0.1 mg/mL) was applied to PSBP conjugated LSPR chip, no absorbance increase was detected (data not shown).

To examine a stable conjugation of PSBP on 5 nm Au NPs, gel electrophoresis analysis was conducted to see if the thiol group (-SH) of PSBP was tethered on the 5 nm Au NPs successfully. Agarose gel (1.0 wt %) was prepared and casted using 1× sodium borate (SB) buffer. When the Au NPs suspension is placed under an electric field in the buffer, it could move according to state of surface charge, which was resulted in the migration of the 5 nm Au NPs as shown in Figure 3a. They show images of agarose gel electrophoresis result of (1) pristine 5 nm Au NPs, (2) PSBP tethered 5 nm Au NPs, respectively.

For the pristine Au NPs, no net charges on the surface of Au NPs resulted in no movement from the well as shown lane (1) of Figure 3a. As the PSBP is negatively charged, the PSBP tethered 5 nm Au NPs could run toward the positive direction under applied voltage as shown in lane (2) of Figure 3a. Therefore, the distance difference between the pristine Au NPs and the PSBP conjugated Au NPs represents distinct evidence that PSBP was concretely conjugated on the 5 nm Au NPs surface.

Figure 3b shows LSPR detection of microplastics using sandwiched PSBP-conjugated 5 nm Au NPs. Again, high concentrations of 1 mg/mL PS samples showed lower absorbance than 0.1 mg/mL as like as Figure 2.

Figure 3c shows a linearity plot of microplastic concentration (0, 0.001, 0.01, 0.1, and 1 mg/mL) vs. LSPR absorbance. The LSPR absorbance were replotted of maximum values from data of Figure 2 and Figure 3b. As shown in Figure 3c, the maximum absorbance of 1 mg/mL was obviously deviated from the linearity. The difference between the absorbance (1.519) calculated from the linear plot and measured absorbance (0.424) of 1 mg/mL was statistically paired t-tested with 5% significance probability. Pearson correlation coefficient (PCC) between the linear plot and the real absorbance at 1 mg/mL was extremely low as 0.00275, where paired two perfect linearities will show 1.0 as the PCC value. Therefore, effective sensing range is from control up to 0.1 mg/mL PS sample. Alternatively, considering the sizes of the oligo-peptide and nanoplastics, the PSBP may have bind to multiple sites of the surfaces of PS fragments. As shown in Figure 3c, the highest increase in absorbance was approximately 0.495 in magnitude. It occurred when the concentration of PS was set to 0.1 mg/mL. It can be seen that its sensitivity has improved up to 60% compared to non-sandwich assay case as shown in Figure 2. Figure 3b shows results of PS fragmented sample smaller than 10 μm through cyro-grinding and meshing.

Separately, it can be seen in Appendix A that the absorbance did not increase when the PS concentration was increased even beyond 0.1 mg/mL with differently sized PS sample. As the PS samples were prepared by employing a sieving process to harvest 75~106 μm PS fragments in Appendix A, most of the PS samples were micro-sized. However, nano-sized debris were believed to be present, as small absorbance increase could be still identified with the 75~106 μm PS fragments.

Appendix A shows UV-Vis absorption spectra to detect 75~106 μm PS fragment with PSBP coupled 5 nm Au NPs sandwich binding, which show higher absorbance increase than that of non-sandwich measurement of Appendix A. As shown in Appendix A, the microplastics, which are relatively huge compared to nanoplastics, did not induce significant LSPR effect [32]. On the contradictory, the microplastics may have scattered UV-Vis incident light and would reduce UV-Vis absorbance regardless of existence of sandwich-bound 5 nm Au NPs. Thus, it is believed that the LSPR sensing could not be observed for the high concentration samples with 75~106 μm PS samples. Instead, the presence of microplastics in high concentrations scattered the incident light inside of the UV-Vis spectrophotometer.

This indicates better nanoplastic detection sensitivity would be available for samples with plastic fragments smaller than 1000 nm in size upon this LSPR system. Microplastics (~5 mm) could screen the surface of the LSPR sensor, especially in high concentration sample. Indeed, the most dangerous microplastics have been reported to be smaller than 100 nm [8,9]. Therefore, LSPR sensors could be more valuable to detect PS nanoplastics in the size range.

Appendix A shows the dynamic light scattering (DLS) results for each PS concentration solution, which shows the number of nanoparticles in the solution according to size. Particle size was analyzed for three times for each concentration of 0.01, 0.1, 1.0, and 10 mg/mL using DLS, and the mean values were 362.37, 191.33, 231, and 514 nm, respectively. As shown Appendix A, there was an optical sample concentration (0.1 mg/mL) to be dispersed well in DI water, which can be kept as nano-sized PS fragments. This confirms that smaller nanoparticles result in better detection efficiency toward LSPR. Regarding the concentrations, the PS fragments could be aggregated to form large particles, which consequently might reduce the LSPR effect.

Figure 4 shows corresponding zeta potentials from DLS results of Appendix A for each PS concentration solution. As shown in Figure 4, an increase in the absolute value of the zeta potential corresponded to an increase in the aggregation resistance between the nanoplastics, which increased the stability of the particles in the solution. Conversely, as the zeta potential approached zero, the particles would become easier to collect or aggregate with each other.

The zeta potentials were measured three times for each concentration of 0.01, 0.1, 1.0, and 10 mg/mL, with the mean values of –39.8, –60.1, –52.8, and –49.277, respectively. As shown in Figure 2 and Figure 3, when the fragmented PS concentration was reached at 0.1 mg/mL, the absorbance variation was maximized; accordingly, the absolute value of the zeta potential of the 0.1 mg/mL PS solution was the highest.

The greater the absolute value of zeta potential is, the more stable nanoplastics can exist. Therefore, the most stable sample concentration was, 0.1 mg/mL, and dispersion of nanoplastics because more unstable it is at 0.01 mg/mL PS. Compared to the previous UV-Vis spectra, 0.1 mg/mL had the highest absorbance and 0.01 mg/mL had the lowest absorbance for the LSPR detection. It critically proves that dispersion of nanoplastics was determinant for the efficient LSPR sensing. A smaller particle size was found to correspond to a larger zeta potential absolute value, and thus better LSPR sensitivity [33].

Figure 5 shows a sensorgram for the sandwich assay implemented for the detection of PS nanoplastics of the 0.1 mg/mL sample [28]. It represents a LSPR time-dependency of PS nanoplastic detection system. Based on the results shown in Figure 3, the absorbance of the 0.1 mg/mL PS solution was selected for time-dependent analysis of the sandwich assay’s ability to directly detect PS particles.

As shown in Figure 5, when the sensor chip was conjugated using a PSBP, the nanoplastic fragments attached to the sensor chip, resulting in small increase in the absorbance at 530 nm, as shown Figure 5. Additionally, the nanoplastics in the 0.1 mg/mL PS solution, which was associated with the largest increase in absorbance (Figure 3), induced LSPR, which consequently altered the refractive index.

As shown in Figure 5, after introduction of sandwich 5 nm Au NPs solution, the absorbance become maximum up to 1.321 in magnitude. However, after DI rinsing at the final step of the sensorgram, the final absorbance was 0.645, which clearly proved stable consecutive LSPR effect by sandwich bound 5 nm Au NPs.

As shown in Figure 5, when nanoplastic sample is ready to be well dispersed, detection time can be very rapid less than 30 min in the LSPR sensing.

One of the time-consuming steps in conventional methods of microplastic or nanoplastic analysis would be the manual differentiation between plastic particles and other interfering particles. Time required for this process can be dependent the separation and purification steps. In addition, manual identification can result in some plastic particles, especially small, transparent particles, being overlooked. Therefore, the LSPR detection system in this study can provide a rapid protocol for nano-size targeted, and transparent sample.

The sustainable existence of PS was also examined by dipping particles in DI water to determine how long PS particles were kept as physically attached state, and whether they would easily detach or not. There was a sudden initial tendency to increase in absorbance, but overall decrease was detected. Consequently, it was confirmed that the PS nanoplastics did not detach out of the specific PSBP binding.

Figure 6 presents high-resolution scanning electron microscope (HR-SEM) surface images of the LSPR chip with high magnification. Unlike the SEM images of chip shown in Appendix A, a platinum coat was applied to observe individual Au NPs. Consequently, individual 40~50 nm Au NPs could be observed. At first, 50,000× magnification image revealed a distinct monolayer of 40~50 nm Au NPs. The massive objects in Figure 6 were expanded to enable identification of nanoparticles smaller than 1000 nm that could induce LSPR effects.

In addition, the FE-SEM image in Figure 6 could not confirm that nanoplastic specific 5 nm Au NPs were firmly bound to the PS nanoplastics sandwiched on the sensor surface due to possible intensive e-beam from SEM measurement.

Appendix A also shows that FE-SEM image of sandwich-bound 75~106 μm PS fragments are denoted by red circles, whereas the non-sandwiched sites associated with 5 nm Au NPs are denoted by green circles. It can highlight larger sized individual Au NPs with circles than unbound probe Au NPs after solely binding of nanoplastic-conjugated 5 nm Au NPs. The circles also indicate that some of Au NPs on the sensor chip were actually bound to the nanoplastics, which were again conjugated by 5 nm Au NPs.

The LSPR detection method was examined with an environmentally available sample in order to check if the same results were accomplished through the real sample. PS is often used as a packing or container material for food, such as noodles in a form of styrofoam. Among the most common products, a cup noodle container which is made of polystyrene could be adopted [33].

In order to examine whether any detectable PS nanoplastics will be generated from a styrofoam container, DI water was poured to a styrofoam container and the DI water was microwaved for 10 min with 700 W power. Then, the water could be boiled and cooled down to room temperature.

Figure 7 shows that styrofoam-derived PS nanoplastics were detected from the microwave boiled DI water. As shown in Figure 7a, more sensitive detection was accomplished for the sandwich assay than non-sandwiched assay as well. In addition, to separate and collect the dissolved PS nanoplastics from the microwave-boiled DI water, PDMS based microfluidic module was adopted as shown in Figure 7b. The measurement of Figure 7b was performed by non-sandwiched method.

The grooved structure was designed to decrease the entropy of PS nanoplastics by way of the hydrophoresis phenomenon [34]. Even more, the microfluidic format was operated as recirculation mode to accelerate concentrating of PS nanoplastics in grooves or traps. As shown in Figure 7b, a high absorption peak under non-sandwiched assay was obtained with the sample concentrated by the PDMS microfluidics, which proves the effect of nanoplastics separation [35]. Based on the linear plot from Figure 2 and Figure 3c for the non-sandwiched assay, the concentration of the microwave-boiled styrofoam sample corresponds to 0.0732 mg/mL and that of the microfluidic concentrated microwave-boiled styrofoam sample corresponds to 0.0934 mg/mL. To identify the existence of PS nanoparticles in the microfluidic concentrated PS sample, the sample solution (100 μL) was air-dried on glass substrate and its image was analyzed by bench-top SEM. Appendix A shows SEM image of the microfluidic concentrated microwave-boiled or extracted PS particles dried on glass. As shown in Appendix A, significant amounts of nanoparticles including micro-sized particles could be identified.

Although microwaving for 10 min created a harsh condition, an apparent detection of nanoplastics was identified, which was a warning of the detrimental effect. The sample liquid preparation from the microwave-boiled DI water was carried out in the similar way as in Reference [33].

In addition, we plan to pretreat a real sample from a Korean river with HCl, to broaden the capability of the LSPR platform.

## 4. Conclusions

In this study, Au NP-layered customized sensor chip for the efficient detection of nanoplastics was characterized using an LSPR-based direct detection system. Specifically, the LSPR sensitivity was increased by employing a sandwich method that attaches 5 nm Au NPs on multiple sides of the target nanoplastic fragments. For practical sample examination on an LSPR chip, microwave-boiled DI water on a styrofoam container was tested with recirculation to show successful detection of the PS nanoplastics in a PDMS nanoplastic-collecting microfluidic module. This system could be developed as an alternative nanoplastic detection system that utilizes conventional UV-Vis spectrophotometer measurements more effectively than systems that employ the conventional direct method of micro/nanoplastic detection.

## Figures and Tables

**Figure 1 nanomaterials-11-02887-f001:**
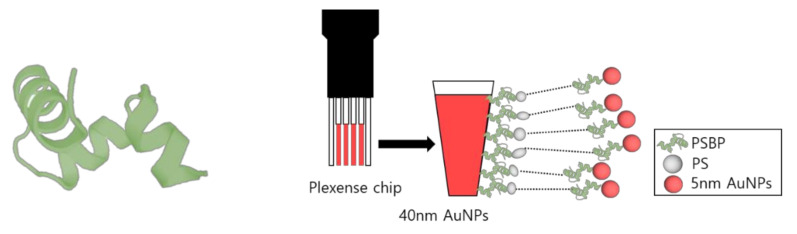
Images of 3D structure of polystyrene-binding peptide (PSBP) and schematic diagram of LSPR sensor chip implemented in sandwich assay method.

**Figure 2 nanomaterials-11-02887-f002:**
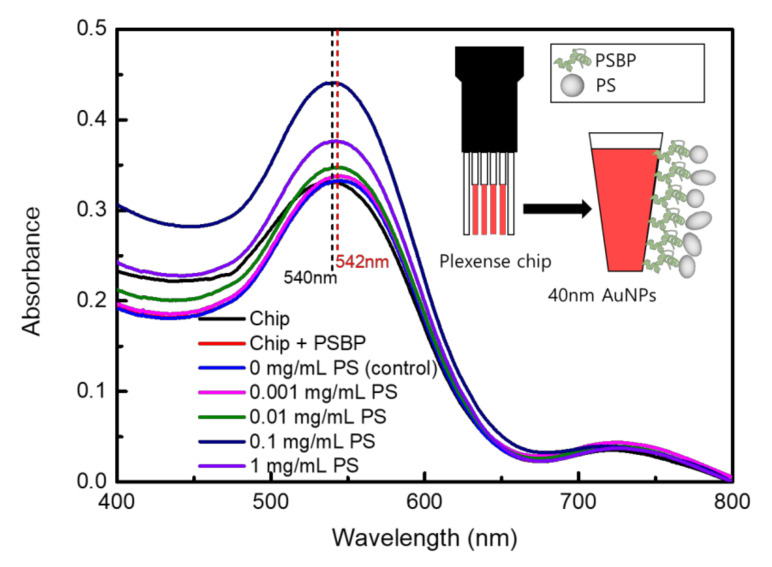
Schematic diagram of LSPR sensor chip that directly attaches to sample microplastics (inlet diagram), and UV-Vis spectrophotometer spectra results of non-sandwich type LSPR detection.

**Figure 3 nanomaterials-11-02887-f003:**
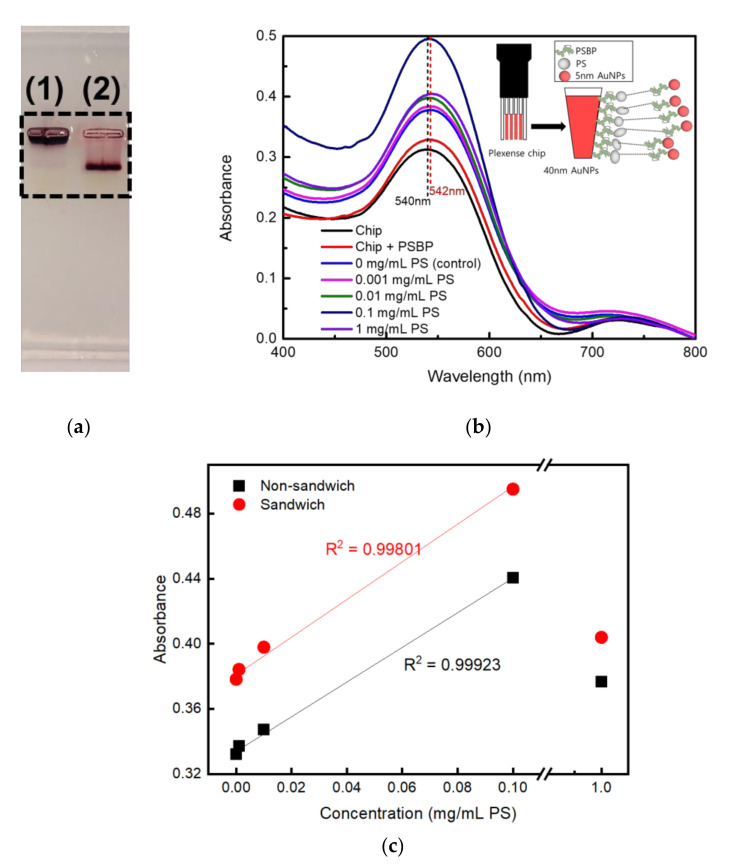
(**a**) Gel electrophoresis analysis of lane (1) 5 nm Au NPs and lane (2) PSBP conjugated 5 nm Au NPs in 1.0 wt % agarose gel, (**b**) UV-Vis measurement results obtained via sandwich assay, and (**c**) linearity plot between maximum absorbance and PS concentration (0, 0.001, 0.01, 0.1, and 1 mg/mL) on LSPR detection.

**Figure 4 nanomaterials-11-02887-f004:**
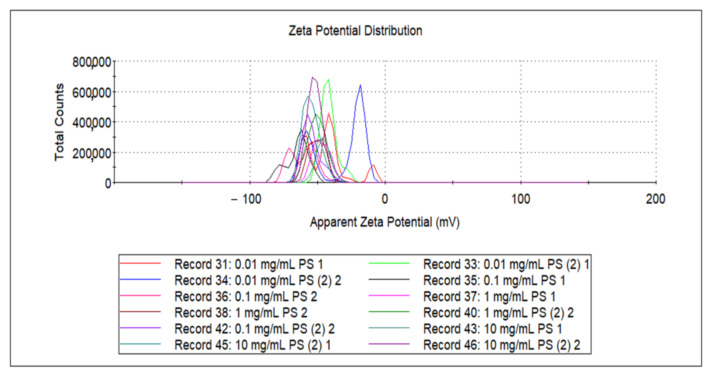
Zeta potential of PS samples at different concentrations.

**Figure 5 nanomaterials-11-02887-f005:**
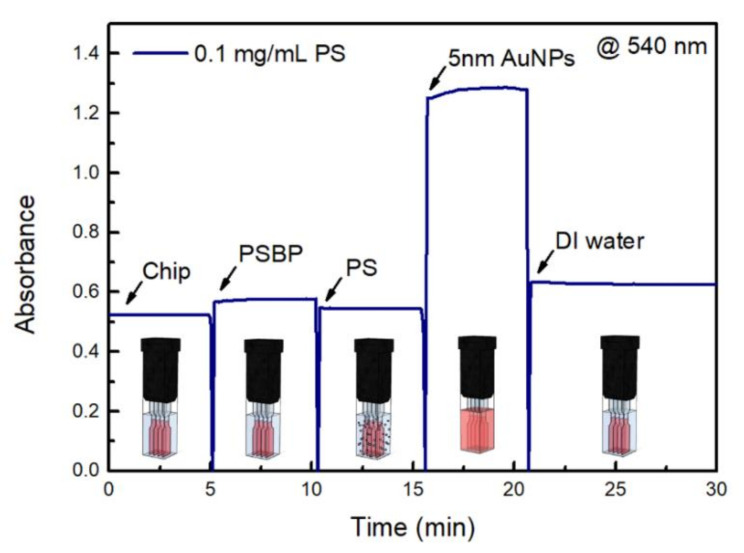
Sensorgram results for sandwich assay UV-Vis spectroscopy-based detection of PS.

**Figure 6 nanomaterials-11-02887-f006:**
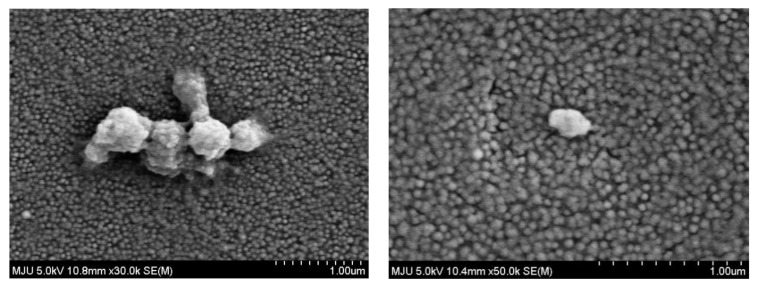
High resolution scanning electron microscope (HR-SEM/EDS) results for the sandwich assay sensor chip with 5 nm Au NPs.

**Figure 7 nanomaterials-11-02887-f007:**
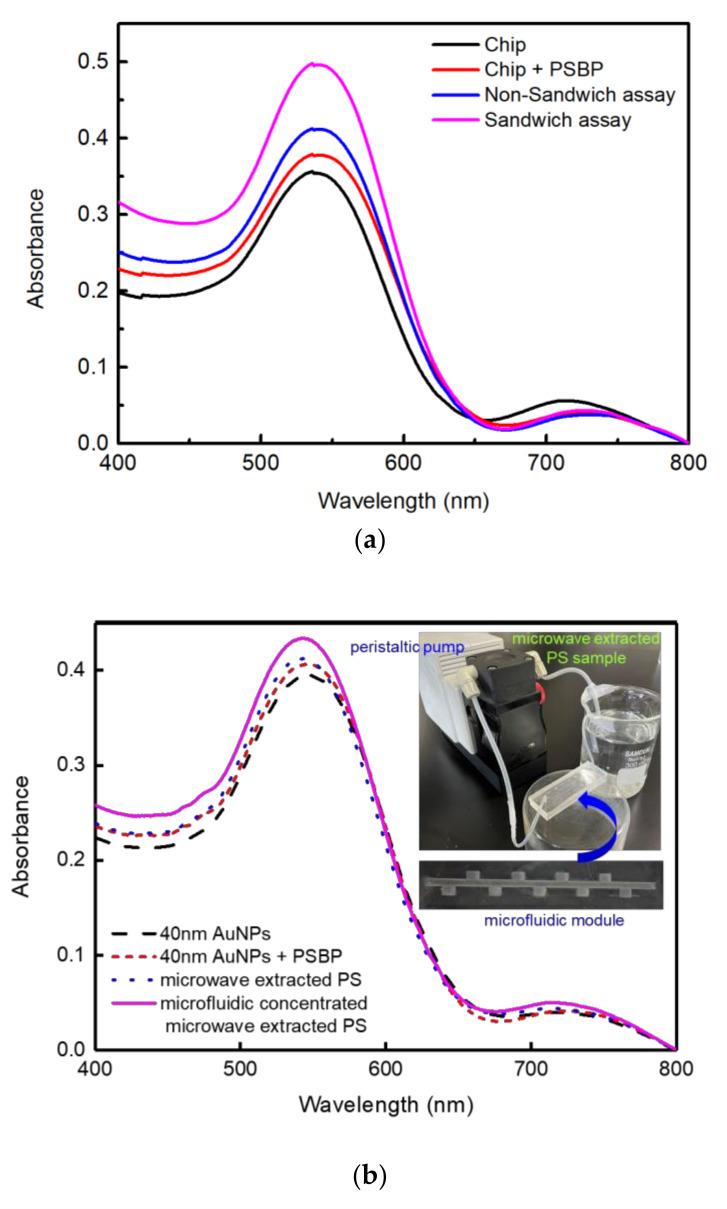
UV-Vis absorption spectra with microwave-boiled DI water in styrofoam bowl for the sandwich assay sensor chip with 5 nm Au NPs: (**a**) pristine styrofoam-derived nanoplastic sample, (**b**) concentrated styrofoam derived nanoplastics in PDMS microfluidic trap module.

## Data Availability

Not applicable.

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
