# Peer review of "Peptide Specific Nanoplastic Detection Based on Sandwich Typed Localized Surface Plasmon Resonance"

_nanomaterials, 2021, doi:10.3390/nano11112887_

Round 1
Reviewer 1 Report
Authors have significanly improved the manuscript.
Author Response
Reviewer #1 :
Authors have significantly improved the manuscript.
Response: Authors would like to thank reviewer for a comment of encouragement.

Reviewer 2 Report
After reviewing the resubmitted manuscript, I believe that the paper has been revised properly according to the suggestions and comments of the reviewer. I, therefore, suggest that this paper should be accepted for publication.
Author Response
Reviewer #2 :
After reviewing the resubmitted manuscript, I believe that the paper has been revised properly according to the suggestions and comments of the reviewer. I, therefore, suggest that this paper should be accepted for publication.
Response: Authors would like to thank reviewer for a comment of encouragement.

Reviewer 3 Report
This paper is proposing a new way to detect microplastics using and LSPR sandwich assay. This is an important area of research. Although the findings of this paper are interesting the following questions should be addressed:
- It was stated in lines 240-243 that since the absorbance change is indistinguishable from the control sample the LOD is 0.001 mg/mL. According to Skoog et. al. in The Fundamentals of Analytical Chemistry text book, the detection limit should be calculated using the formula ksb/m, with k = 2 or 3, sb = standard deviation of the blank, and m= the slope of the calibration curve. This equation accounts for noise in the blank.
- In lines 286 to 290 , the range is claimed to be 0 to 1 mg/mL by saying that 1 mg/ml “..obviously deviated from linearity.” This value should be quantified by seeing where the data deviates by 5 % from the line using a limit of linearity calculation.
- In lines 366-368, the author’s claims that “…absorbance at 540 nm was not enhanced effectively due to the huge size of the 75-106 um fragmented PS.” However, the absorbance does show an absorbance change. The authors should clarify what value of change is considered enhanced and show data that supports that claim.
- In Figure 7 it would be good to good to quantify the change in absorbance seen based on the calibration curve. Why does Figure 7b have less of an increase in absorbance when microfluidics concentrates the sample? In addition, and SEM of the sample would be helpful and would strengthen the analysis.
Author Response
Reviewer #3 :
This paper is proposing a new way to detect microplastics using and LSPR sandwich assay. This is an important area of research. Although the findings of this paper are interesting the following questions should be addressed:
Response: Authors would like to thank reviewer for kind comments. And, based on the comments, many ambiguities in the manuscript could be eliminated.
It was stated in lines 240-243 that since the absorbance change is indistinguishable from the control sample the LOD is 0.001 mg/mL. According to Skoog et. al. in The Fundamentals of Analytical Chemistry textbook, the detection limit should be calculated using the formula ksb/m, with k = 2 or 3, sb = standard deviation of the blank, and m= the slope of the calibration curve. This equation accounts for noise in the blank.
Response: Authors are very grateful about this valuable comment. Obviously, authors did not recognize an exact definition of the LOD in an academic standard. So, in this paper, the LOD was not obtained by the definition, from the formula with statistical error-based measurement with a blank sample. Therefore, the manuscript was corrected by replacing the terms of LOD with limit of quantification (LOQ), which can be defined a lowest concentration of a substance that is possible to be determined by the LSPR of this study. Authors would like to thank reviewer again for this comment.
Authors added the lines of 244-247.
“Therefore, it was defined that a limit of detection quantification (LOQ) was 0.001 mg/mL PS, which could be defined as a lowest concentration of a substance that is possible to be determined by the LSPR of this study.”
In lines 286 to 290, the range is claimed to be 0 to 1 mg/mL by saying that 1 mg/mL“..obviously deviated from linearity.” This value should be quantified by seeing where the data deviates by 5 % from the line using a limit of linearity calculation.
Response: Authors would like to thank reviewer for helpful comments. The difference between the absorbance (1.513) calculated from the linear plot and measured absorbance (0.424) of 1 mg/mL was statistically paired t-tested with 5% significance probability. Pearson correlation coefficient (PCC) between the linear plot and the real absorbance at 1 mg/mL was extremely low as 0.002745, where two perfect linear matches will show as 1 as the PCC value.
Authors added the lines of 293-298.
“The difference between the absorbance (1.519) calculated from the linear plot and measured absorbance (0.424) of 1 mg/mL was statistically paired t-tested with 5% significance probability. Pearson correlation coefficient (PCC) between the linear plot and the real absorbance at 1 mg/mL was extremely low as 0.002745, where paired two perfect linearities will show 1.0 as the PCC value.”
In lines 366-368, the author’s claims that “…absorbance at 540 nm was not enhanced effectively due to the huge size of the 75-106 um fragmented PS.” However, the absorbance does show an absorbance change. The authors should clarify what value of change is considered enhanced and show data that supports that claim.
Response: Authors would like to thank reviewer for critical comment. The author’s claims in lines 366-368 were not enough to clarify change of the enhancement of sensing. Therefore, authors removed the lines 366-368 in the manuscript and removed Figure S6 of the sensorgram. Therefore, the Figure S6 in new manuscript of supplementary information material becomes “FE-SEM measurement results for the sandwich assay of 75~106 mm PS fragments”
In Figure 7, it would be good to quantify the change in absorbance seen based on the calibration curve. Why does Figure 7b have less of an increase in absorbance when microfluidics concentrates the sample? In addition, and SEM of the sample would be helpful and would strengthen the analysis.
Response: Authors would like to thank reviewer for helpful comments. Based on the calibration linear plot (not including 1mg/mL PS sample) between absorbance and PS concentrations (75~106 mm PS fragment-based concentration) shown in Figure 2 and Figure 3(c), the estimated concentration of the styrofoam microwaved sample was 0.0732 mg/mL in Fig. 7(a). With the PDMS microfluidic module, the concentrated microwaved PS sample shows 0.0934 mg/mL. In addition, Fig. 7(b) was absorbance profile with non-sandwiched method, where the sandwiched method could not be performed smoothly along with the PDMS microfluidic trap module. It is why Figure 7(b) has the less increased absorbance than the absorbance of sandwiched assay shown in Figure 7(a).
Authors added the lines of 426-427 and 435-440.
“The measurement of Fig. 7(b) was performed by non-sandwiched method.”
“As shown in Fig. 7(b), a high absorption peak under non-sandwiched method was obtained with the sample concentrated by the PDMS microfluidics, which proves the effect of nanoplastics separation [35]. Based on the linear plot from Fig. 2 and Fig. 3(c) for the non-sandwiched assay, the concentration of the microwave-boiled styrofoam sample corresponds to 0.0732 mg/mL and that of the microfluidic concentrated microwave-boiled styrofoam sample corresponds to 0.0934 mg/mL.”
During second revision, authors have tried to have SEM images with the microplastics from the styrofoam microwave-extracted samples. However, since the electron beams were too strong with HR-SEM, the nano-sized PS or organics were rapidly melted out, which could not have a clear image. However, we could get an image with microplastics from a low resolution SEM on bare glass, which was coated with Au.
Authors corrected and added the lines of 151-157 and 440-445.
“For the scanning electron microscopy (SEM) analysis, using a field emission scanning electron microscopy (FE-SEM, SU-70, Hi-tachi Co., Osaka, Japan), surface of LSPR chip was analyzed. Because the LSPR chip was a nonconductive target, Au was used as a coating layer to enable SEM measurement. In addition, from a bench-top SEM (COXEM, EM-30AX, Korea), microplasticts dried on glass surface could be imaged with Au coating.”
“To identify the existence of PS nanoparticles in the microfluidic concentrated PS sample, the sample solution (100 mL) was air-dried on glass substrate and its image was analyzed by bench-top SEM. Figure S7 shows SEM image of the microfluidic concentrated microwave-boiled or extracted PS particles dried on glass. As shown in Fig. S7, significant amounts of nanoparticles including micro-sized particles could be identified.”

Round 2
Reviewer 3 Report
The authors have sufficiently addressed the concerns
This manuscript is a resubmission of an earlier submission. The following is a list of the peer review reports and author responses from that submission.
Round 1
Reviewer 1 Report
Oh et al. described a method for AuNP-based sandwich detection of nanoplastics via an oligopeptide recognition scheme. Briefly, an LSPR sensor comprising of a AuNP monolayer-coated commercial plastic sensor is functionalized with an oligopeptide probe that selectively binds to polystyrene (PS), referred to as polystyrene-binding peptide (PSBP). After PS nanoplastics are captured on the sensor surface, 5 nm AuNPs, which are likewise pre-functionalized with PSBP, were added to form a sandwich assay. The amount of nanoplastics bound to the sensor were mainly inferred from absorbance changes of the LSPR peak, measured via UV-vis spectroscopy. There are a few issues that the authors need to address before further review.
- First and foremost, there needs to be a clear rationale behind the use of AuNPs with such dimensions. Specifically, why 40-50 nm AuNP on the monolayer and 5 nm AuNP for the sandwiching layer (e.g., why not 20-30 nm AuNP on the monolayer and 15 nm AuNP for the sandwiching layer). Small differences in AuNP dimensions can lead to significant differences in terms of LSPR sensing performance, due to the electromagnetic enhancement effects. Therefore, the rationale needs to be explained in detail; if not, the optimization experiments need to be performed and included in the supporting information.
- In most LSPR sensing setups, the detection event is detected based on peak shifts rather than absorbance intensity changes. However, in this case, the authors seem to rely mostly on absorbance intensity changes, which suffer from a couple of severe limitations (as the authors themselves have pointed out). In particular, the absorbance intensity is significantly diminished in the presence of micro-sized plastics. The authors need to carefully rethink the sensing approach and decide on which LSPR sensing modality (i.e., absorbance-based, absolute peak shift-based, extinction difference-based, centroid peak-based) is optimal to track the binding event.
- Once the sensing modality is revised, the authors also need to include a dose-response curve to determine the limit-of-detection, sensing response sensitivity, linear range, limit of linearity, etc. for sensing of the PS nanoplastics.
- I believe there are also various other types of micro/nanoplastics besides PS that needs to be detected in the real settings, please include a discussion on how the authors can broaden the sensing capability of the platfom.
- As the authors have pointed out, real plastic waste samples exist in various shapes and dimensions and to specifically mention the detection of nanoplastics seem to be narrowing the scope too much. Firstly, how can end-users of this platform differentiate or sieve out microplastics from nanoplastics? Secondly, is it possible for the platform to detect both microplastics and nanoplastics using the same sensing modality?
- From the visual evidence from HR-SEM in Figure 6, it seems that the 5nm PSBP-functionalized AuNPs tend to aggregate around the bound nanoplastic. Is this type of aggregation specifically due to the binding on the nanoplastic or does the aggregation occur spontaneously in solution? The authors need to check this.
Reviewer 2 Report
This work attempts to detect PS nanoplastics by using a sandwich assay with 40~50 nm Au NPs and 5 nm Au NPs and a peptide probe. In general, the conclusions are not quite supported by the results. The following major points should be considered:
- The detection system should be better described. Since the most of the PS particles are in micron sizes, light scattering effect is very serious, how do the detection system or the method eliminates the light scattering effect to demonstrate the method is good for PS nanoplastics?
- Why the bottom 40~50 nm Au NPs layer is necessary in this study? Instead, with the peptide probe modified on a bare substrate and using the 5 nm Au NPs to follow the sandwich assay may even be more sensitive.
- The selectivity of the method to PS against other kinds of plastics has not been demonstrated.
As examples, other more specific points are as following:
- What is meant by dual plasmonic or LSPR effect?
- How to prepare 5 nm AuNP? Why it is no charge? Most Au NPs have surface charges.
- Page 9 and Figure 5, in the 3rd step after adding PS, the absorbance decreases, why? Instead, it should increase.
- The resolution of Figure 6 is not able to see the 5 nm Au NPs.
Reviewer 3 Report
The manuscript « Peptide specific nanoplastic detection based on sandwich typed localized surface plasmon resonance” by Oh et al, presents the possibility to detect micro plastic using specific peptides.
The importance of the subject is high but the manuscript does not present results demonstration the analytical method. No control is presented.
It is necessaire to show the calibration curve of the detection and to determine analytical parameters of the methods (limit of detection, dynamic range, reproducibility, specificity…).
Negative controls using other micro materials should be include in the study to show the specificity of detection. It is also important to show that sandwich between AnNPs of 40 nm and AuNPs of 5 nm cannot be formed in the absence of PS micro plastics.
Reviewer 4 Report
The paper describes an interesting work on LSPR-based sensing of nanoplastic fragments by using PS specific oligo-peptide attached to the AuNP layer. The authors also demonstrated the enhancement of sensitivity by sandwich binding of smaller AuNPs. It is highly appreciated that the authors chose nanoplastics fragmented ground microplastics, aiming to practical use. The significance of the study is clearly stated and the results are well presented and discussed. I consider this work suitable for publication in Nanomaterials after the authors address the points below:
1) The authors should describe the method for the preparation of PS nanoplastics more clearly in detail. Were 10 um sized PS and 75-106 um sized PS particles ground? How were the particles prepared?
2) What is the difference between two lanes of electrophoresis presented in Fig. 3(b)?
3) Is there any direct or systematic relationship between PS concentration and the value of zeta potential? Is it coincidence or inevitable that the zeta potential of 0.1 mg/mL PS solution was the highest?
4) The second paragraph on page 10 is somewhat confusing. Are 5 nm AuNPs visible in the SEM image? How did the authors confirm the presence of 5 nm AuNPs? What is the meaning of “plasmonic coupling effects” in the last sentence? The coupling between 5 nm AuNPs, or between a 5nm AuNP and a 40-50 nm AuNP?